# Manipulation of Jasmonate Signaling by Plant Viruses and Their Insect Vectors

**DOI:** 10.3390/v12020148

**Published:** 2020-01-27

**Authors:** Xiujuan Wu, Jian Ye

**Affiliations:** 1State Key Laboratory of Plant Genomics, Institute of Microbiology, Chinese Academy of Sciences, Beijing 100101, China; wuxj@im.ac.cn; 2CAS Center for Excellence in Biotic Interactions, University of Chinese Academy of Sciences, Beijing 100049, China

**Keywords:** plant virus, insect vector, convergent evolution, JAZ-MYC hub

## Abstract

Plant viruses pose serious threats to stable crop yield. The majority of them are transmitted by insects, which cause secondary damage to the plant host from the herbivore-vector’s infestation. What is worse, a successful plant virus evolves multiple strategies to manipulate host defenses to promote the population of the insect vector and thereby furthers the disease pandemic. Jasmonate (JA) and its derivatives (JAs) are lipid-based phytohormones with similar structures to animal prostaglandins, conferring plant defenses against various biotic and abiotic challenges, especially pathogens and herbivores. For survival, plant viruses and herbivores have evolved strategies to convergently target JA signaling. Here, we review the roles of JA signaling in the tripartite interactions among plant, virus, and insect vectors, with a focus on the molecular and biochemical mechanisms that drive vector-borne plant viral diseases. This knowledge is essential for the further design and development of effective strategies to protect viral damages, thereby increasing crop yield and food security.

## 1. Introduction

There are many plant viruses in natural and agricultural ecosystems. The majority of plant viruses are transmitted by piercing-sucking insects. Plant-mediated interactions between viruses and insect vectors greatly influence the population dynamics of the vectors and plant disease epidemiology.

Around 70 years ago, the discovery of the phenomenon that aphids (*Aphis fabae*) benefit from feeding on virus-infected leaves initiated a new research area in plant virology and the tripartite interaction of virus–vector–plant [1]. Since then, increasing evidence has clearly demonstrated that insect-borne plant viruses generally have effects on their vectors directly and/or indirectly. Many persistently transmitted viruses have direct effects on their insect vectors including biting rates, feeding amount and the immunity alterations of defensive related gene expression, as reviewed previously [2,3,4,5]. Furthermore, several insect-borne plant viruses have been shown to indirectly manipulate the behaviors of their vectors to promote their own transmission. Firstly, viruses influence the feeding preferences of their vectors. These persistent transmitted viruses, such as geminiviruses, bunyaviruses and luteoviruses, often increase the attraction of their vectors to virus-infected plants [6,7,8,9]. Meanwhile, some nonpersistent transmitted viruses, such as bromoviruses and potyviruses, can take a pull-push strategy in their aphid vector to optimize the transmission, which indicates an initial settlement on infected plants and a later preference for mock-inoculated plants [10]. Secondly, feeding activity of vectors also changes in virus-infected plants. For example, *Aphis gossypii* spends a longer time feeding from the phloem in virus-infected plants than in non-infected plants, thus increasing the probability of virus acquisition and inoculation [11]. Thirdly, parallel to the short-term impact on vector behaviors, viruses can alter the plant host’s metabolic profile to affect the fitness of their vectors [12,13,14]. Viruses enhance the performance of their vectors by repressing anti-herbivore defensive chemicals like indole and aliphatic glucosinolates (GSs), or by improving nutrition (such as amino acids) in infected plants [15,16].

Due to the rapid improvement in breakthrough technologies being developed in genomics and genome editing, the mechanistic understanding of these tripartite interactions has been much improved over the recent decade. Jasmonate (JA) signaling tends to be a convergent target manipulated by plant viruses and herbivores. As one of the most critical defensive phytohormones, JA is often involved in a plant’s defense against pathogens (especially necrotrophic pathogens) and herbivores [17,18,19]. Additionally, several studies have supported the concept that JA is critical for antiviral defense [20,21,22,23]. In this review, we have summarized how JAs participate in the tripartite interactions between plants, vector-borne viruses and herbivores. We have also highlighted how viruses and the insect vector hijack plant JA-mediated defenses to promote the insect population and viral transmission. 

## 2. Overview of Jasmonate Derivatives (JAs) and JA Signaling 

Due to the essential role of JA signaling in regulating plant defense, the JA biosynthesis, signal amplification and feed-back repression mechanisms have frequently been the focus of research and reviews [24,25,26]. JAs have been well known as defensive phytohormones that respond to unfit growth conditions and other kinds of abiotic and biotic challenges. Some JAs can also be released to function as communication signals between plants in anticipation of mutual dangers [27,28]. In most cases, JAs function as chemical triggers to induce the biosynthesis of various defensive chemicals and proteins to counter pathogens and herbivores. Plant pattern-recognition receptors (PRRs) perceive molecular patterns from pathogens and herbivores to recognize herbivore-associated molecular patterns (HAMPs), damage-associated molecular patterns (DAMPs), and microbe/pathogen-associated molecular patterns (MAMPs/PAMPs) and thereby initiate JA signaling-dependent resistance [29,30]. These plant elicitor molecules can be peptides such as systemin, extracellular ATP (eATP), sucrose, volicitin, fatty acid conjugates, caeliferins and others [31,32]. Following the perception of damage signals and cellular calcium flux, the activation of mitogen-activated protein kinase (MAPK) cascades amplifies the signaling from local to systemic leaves and also from the cytosol into the nucleus, where JA and its derivatives function to transcriptionally reprogram sets of defensive gene expression.

The activation of JA signaling mainly depends on the regulation of the F-box protein CORONATINE INSENSITIVE1 (COI1), which functions as a receptor of JA-Ile, together with JASMONATE ZIM (JAZ) repressor proteins, in the E3 ubiquitin-ligase SKP1-Cullin-F-box complex [33]. JAZs negatively regulate the JA-signaling pathway. In resting cells, JAZ repressor proteins together with the adaptor protein NOVEL INTERACTOR OF JAZ (NINJA) and the recruited corepressor TOPLESS (TPL) bind with positive transcriptional factors, such as basic helix-loop-helix MYCs, to inhibit the activation of the JA pathway [34]. In JA-stimulated cells, JAZ are degraded via the proteasome after the jasmonic acid-isoleucine (JA-Ile) signal binds to COI1, leading to the activation of downstream gene expression and immunity [35,36,37]. These JAs-responsive transcription factors belong to different families including bHLH, MYB, AP2/ERF and WRKY [38]. These transcriptional regulators integrate a multilayer defense to external stress. Generally, there are two branches (the MYC branch and the ERF branch) of JA signaling based on the studies of *Arabidopsis thaliana*. One well-studied branch is controlled by the bHLH transcription factors, MYCs. The positive regulation of the synthesis of toxic proteins, such as VEGETATIVE STORAGE PROTEINs (VSPs), mainly induces defensive responses against wounding and attacks by insect herbivores [39,40,41] (Figure 1). The second ERF branch is co-regulated by the phytohormones JA and ethylene (ET) to counter necrotrophic pathogens by inducing the expression of the defense marker gene *PLANT DEFENSIN1.2* (*PDF1.2*) [42]. The APETALA2/ETHYLENE RESPONSE FACTOR (AP2/ERF) family transcription factors function by forming protein complexes, such as ERF1 and OCTADECANOID-RESPONSIVE ARABIDOPSIS59 (ORA59) [43,44].

This sophisticated JA pathway can confer versatile and adaptive traits to plants by fine tuning the balance between defense and development [45]. Defective JA reception will aggravate the symptoms caused by RNA viruses, the nematode herbivory induces JA-mediated stem cell activation and regeneration, suggesting the critical roles of JA in plant developmental regulation [22,46]. Meanwhile, JAs and JA signaling serve multiple roles in response to herbivores by priming direct and indirect defenses of plants [47,48,49,50]. The levels of JA-Ile directly affect the fitness of herbivores [51]. A more effective method is to amplify the JA signal after perceiving an infestation of herbivores. JA signaling mediates various specialized metabolites against herbivorous attackers, such as terpenoids, alkaloids, and GSs [41,52]. For instance, the biosynthesis of toxic GSs metabolites contributes to the defense of both phloem-feeding and chewing insects [40,53]. The ectopic induction of these JA pathway gene expressions and metabolites directly confers biotic stresses. There were several early reports about extraneous JAs application efficiently reducing the infection of DNA or RNA viruses. The foliar application of methyl jasmonate was shown to reduce the *Rice black-streaked dwarf virus* incidence on rice [54]. JA treatment also reduces the accumulation and symptoms of geminiviruses [55]. Several types of volatile organic components (VOCs) can be induced by attacks of pathogens/pests or the direct application of JAs, such as terpenes/terpenoids and green leaf volatiles (GLVs). These inducible volatiles are a vital type of defensive signal in the infested plants [52,56]. Therefore, JAs and JA signaling are critical for plant community resistance to herbivores and pathogens.

## 3. The Counter-Defense of Virus and Herbivore to JA Signaling in Plants

### 3.1. The Herbivore Manipulation of JA Signaling

Although plants are equipped with sophisticated defensive mechanisms, herbivores and pathogens have multiple counter defense strategies adapted to their host plants. The antagonistic relationship between the two defensive phytohormones salicylic acid (SA) and JA, is the common target to manipulation. *Pieris brassicae* can mobilize the plant SA pathway by strongly diminishing one of the key regulators of JA signaling, MYCs, to repress host antiherbivore defenses [57]. Similarly, the infestation of Mealybug (*Phenacoccus solenopsis*) increases SA levels and decreases JA production, thereby reducing antiherbivore defenses [58]. Furthermore, herbivore can exploit symbiotic bacteria to evade host defenses. For instance, symbiotic bacteria (*Pseudomonas* sp.) in potato beetle (*Leptinotarsa decemlineata*) larvae can elicit SA signaling and suppress the JA-responsive plant defenses against herbivore [59].

Moreover, many salivary effectors of herbivores have been identified to help them conquer host defenses [60,61,62]. For instance, the NlEG1 effector in the salivary excretions of the brown planthopper has endoglucanase activity that enables the planthopper to feed on rice [63]. A class of Ca++-binding proteins, which are critical for preventing sieve–plate occlusion, is recognized to be used by phloem-feeding insects [64,65]. This kind of herbivore salivary effector plays a vital role in a plant’s defense modulation, especially to manipulate JA defense through SA. Whitefly has long been found to suppress JA defenses by eliciting SA signaling [66]. The whitefly salivary effector Bt56 could interact with the host NTH202 transcription factor to active SA signaling, thus suppressing effective JA defenses [67]. The oral secretions of beet armyworm (*Spodoptera exigua*) use their glucose oxidase activity to elicit a SA burst and attenuate JA levels [68].

Besides exploiting the indirect antagonistic relationship between SA and JA, the salivary effectors of herbivores can directly target JA signaling components to circumvent plant defenses during feeding. We identified that the whitefly salivary effector Bsp9 (also reported as Bt56) interacts with the JA-regulated WRKY33 transcription factor, which positively regulates plant resistance against whiteflies by inducing the expression of the *PDF1.2* gene [69]. Another effector HARP1 from cotton bollworm oral secretions can interact with JAZ repressors to prevent COI1-mediated JAZ degradation, thus blocking the JA mediated host defense [70]. Downstream antiherbivore chemical toxins can also be the targets of herbivores. Caterpillars (*Helicoverpa zea*) can secrete glucose oxidase to reduce the production of JA-mediated toxic nicotine in tobacco [71]. GSs are the most important defense compounds in Brassicales controlled by JA signaling [72]. When induced with a herbivorous attack, myrosinases convert the GSs substrate to release toxic anti-herbivore chemicals. To counter this host defense, *Bemisia tabaci*, a generalist and phloem-feeding insect, is able to cleave off the sulfate group of intact GSs so that it cannot further function as the substrate of myrosinases [73].

### 3.2. JAZ-MYC is one of the Common Targets of Plant Viruses

Insect vectors can acquire benefits to carry and transmit plant pathogens that help them suppress a plant’s antiherbivore defenses. Besides their well-known roles in pathogen and herbivore defense, JAs and JA signaling mediate antiviral defenses in plants [20,21,22,23]. In this way, viruses have to evolve ways to suppress the JA signaling of a plant to achieve survival by mutation or DNA/RNA recombination. Many viral factors can repress JA-regulated gene expression, such as βC1 of the begomovirus satellite, 2b of the *Cucumber mosaic virus* (CMV) and HC-Pro of the *Turnip mosaic virus* (TuMV) [55,74,75]. There is selective pressure for viruses to manipulate plant defenses to assist pathogen vector transmission. As stated above, JA mediates multiple transcriptional modules in a plant’s antiherbivore defense, especially the MYC branch. The JAZ-MYC hub is one of the key nodes in JA signaling and MYC2-orchestrated transcriptional reprogramming during JA signaling [76]. Notably, during its long evolution history, JAZ-MYC hub has become a common target for both insect-borne plant DNA and RNA viruses [77,78,79,80].

We previously identified the βC1 of the monopartite begomovirus *Tomato yellow leaf curl China virus* (TYLCCNV) as the viral factor that suppresses plant terpene biosynthesis. βC1 directly interacts with MYC2 to subvert its transcriptional activity, compromising the activation of MYC2-regulated terpene synthase genes, and promoting whitefly vector performance. In addition, MYC2 is an evolutionarily conserved target of begomoviruses due to its association with bipartite begomoviral protein BV1 [77]. Another begomovirus lacking the DNA satellites *Tomato yellow leaf curl virus* (TYLCV) also targets the JAZ-MYC hub to attenuate JA defense to promote vector performance. The TYLCV C2 protein interacts with plant ubiquitin to compromise JAZ1 degradation, thus inhibiting MYC2-regulated terpene defense. This strategy was also found to be conserved among non-satellites begomoviruses [78]. Besides plant DNA viruses, we have also shown that viruses from one of the negative-sense RNA orthotospoviruses could manipulate their thrip vector’s preferential behavior by targeting MYC-mediate defense. *Tomato spotted wilt orthotospovirus* (TSWV) encodes nonstructural protein (NSs) to directly interact with MYC2 and its two close homologs MYC3, MYC4, to disable JA-mediated host defenses against the western flower thrip (*Frankliniella occidentalis*) vector. The dysfunction of the MYCs in *Arabidopsis thaliana* leads to enhanced vector attraction and performance. Moreover, the association between MYC2 and NSs is conserved among different orthotospoviruses and plant hosts [79]. In addition, 2b of the *Cucumber mosaic virus* (CMV) also attenuated the JA signaling pathway. 2b directly interacts with and represses the JA-induced degradation of host JAZ proteins, thus inhibiting JA signaling. The *myc234* triple mutant plants were observed to attract the CMV aphid vector [80]. These similar results suggest the general features of JAZ-MYC hub manipulation by viruses in the tripartite interactions of the virus–vector–plant (Figure 2).

Besides plant viruses, the bacterial pathogen *Pseudomonas syringae* delivers effectors like HopX1, HopZ1a, AvrB to depredate JAZs for successful infection [81,82,83,84,85,86,87] (see more information in Table 1). These studies reveal the potential convergence of pathogen manipulation tactics at the MYC-JAZ hub. We may regard this convergence as a consequence of adaptive traits evolved by pathogens to counter plant host defense systems. The JA signaling pathway is highly conserved in land plants. Recent discoveries have clarified that the single MpJAZ in *Marchantia polymorpha* is an ortholog of AtJAZ in Arabidopsis with a conserved function, such as repressing jasmonates biosynthesis, senescence, plant defenses, and promoting cell growth [88]. A previous study analyzed the evolution of more than 1000 JAZ sequence proteins using bioinformatics showing highly conserved features along the evolutionary scale [89]. These findings indicated that the JA-regulated defense pathway was relatively conserved both in higher and lower plants. This highly conserved property being targeted by pathogens or insects seems to be an inevitable result.

If the hypothesis that viruses convergently target the JAZ-MYC hub is true, we must determine the co-infection situation in natural systems, in which diverse viruses, insect vectors, and nonvector herbivores coexist and interact. A simple example of this relationship is the cooccurrence of whitefly and thrip insects and their transmitted TYLCV and TSWV on the same plant. These two persistently transmitted viruses both target the JAZ-MYC2 hub [77,79]. This process is supposed to benefit both vector and nonvector insects since JAZ-MYC2 is one of the key hubs to positively regulate defenses against multiple herbivores. Unexpectedly, it is reported that when whiteflies transmit TYLCV and western flower thrips transmit TSWV, they only benefit their own vectors and restrict the transmission behaviors of the other [90]. The following offer some explanations to resolve this superficial conflict. First, these vectors and nonvectors, including the feeding cell types, have different feeding patterns. It is expected that plants can sense different PAMPs from different types of herbivores and then prime different JAZs-mediated downstream defensive events. Beside the difference in PAMPs’, the feeding cell types and feeding styles are both different for whitefly and thrip. Whiteflies specifically feed on the phloem, while thrips mainly feed on mesophyll tissue. Concerning differences in feeding style, whitefly is a piercing-sucking insect, and thrip is a rasping-sucking insect. JAZ-MYC acts as an integrator to regulate upstream input signals and then activate different transcription modules for specific outputs. Secondly, besides the JAZ-MYC hub, a virus can target multiple host proteins. One well-known example is βC1 of TYLCCNV. βC1 can interact with at least three host plant proteins—AS1, WRKY20, and MYC2—to regulate a plant’s defense against herbivores [77,91,92]. Recent work in our lab showed that a virus could certainly solve this kind of dilemma by targeting multiple host proteins [92]. The begomovirus employs the βC1 protein to reprogram plant immunity to promote the performance of the whitefly vector and suppress the performance of nonvector insects like cotton bollworms and aphids. βC1 hijacks WRKY20, which is notably expressed in the phloem to spatiotemporally redeploy the plant’s chemical immunity within the leaf and asymmetrically affect vectors and nonvector competitors [92].

### 3.3. JA-Regulated Chemical Defense Hijacked by Plant Viruses

As good partners of insect vectors, plant viruses can inhibit the JA-regulated biosynthesis of antiherbivore metabolites [93]. For example, plants produce a series of chemical volatiles to communicate with insects (to repel herbivores or to attract natural enemies). Terpenoids are the most abundant compounds in plant volatiles. Sesquiterpenes and monoterpenes, such as (E)-α-bergamotene, (E)-β-caryophyllene, linalool, etc., are reported to generally have an effect on an insect’s feeding orientation [94,95,96,97]. Many pathogens manipulate these chemical volatiles to alter their vectors’ behaviors, especially insect-borne plant viruses, whose spread is highly reliant on insect vectors [2,98,99]. As reported, (E)-α-bergamotene is a whitefly repellent released by *Nicotiana benthamiana*. TYLCCNV associated with TYLCCNB (*Tomato yellow leaf curl virus* betasatellite) reduce the amount of this compound, thus resulting in an attraction of the insect vector [77]. TSWV also reduces the amount of linalool, whose function is to repel the thrip vector [79].

In addition, as stated before, some plant viruses increase the suitability of their insect vectors by regulating their contents of secondary metabolites in plant cells according to their own transmission characteristics [2]. For instance, for a persistently transmitted virus, virus acquisition by vectors usually take hours; this kind of virus tends to alter the plant to provide better nutrition and ensure fewer defenses to improve the performance of vector insects [100,101]. Generally, this type of virus will shorten a vector’s development period (some vectors also reduce the period of vulnerability to predation) and increase its numbers of offspring [102]. Meanwhile, non-persistently transmitted viruses can be acquired in minutes, and these viruses tend to reduce plant quality to reduce the vector’s residence time, thereby increasing propagation efficiency [14,103].

## 4. The Plant Defense-Growth Trade-Off Regulated by JAs

Increased plant defense usually leads to poor growth [104]. In order to optimize their reproductive success in dynamic environments, plants have evolved a sophisticated mechanism. As a major part in plant defense, the switch between the transcriptional repression and activation of JA signaling regulates a plant’s defense–growth trade-off. The key factors in JA signaling like JAZ proteins prevent unrestrained immune responses to partly promote growth [104]. In addition, some studies suggest that JA fine-tunes the growth-defense dilemma through crosstalk with other signaling pathways. The BRI1-EMS-SUPPRESSOR1 transcription factor controlled by Brassinosteroids was reported to participate in this process by antagonizing JA-activated plant defenses [105]. Moreover, the JAZ and MYC proteins interact with central proteins, such as DELLA, HY5, and IAA59 in other plant growth-related hormone-dependent pathways to fine tune defense and growth [106,107,108,109,110]. For instance, in *Arabidopsis thaliana*, gibberellin (GA) is often related to plant growth regulation. It can de-repress the inhibition of the DELLA protein to the growth regulators PHYTOCHROME INTERACTING FACTORs (PIFs). On the contrary, JA promotes the DELLA proteins to inhibit PIFs and degrades JAZ proteins to activate MYC2, thereby exerting negative effects on growth but positive effects on defense [111,112]. Moreover, plant viruses attenuate the plant defense by targeting the JAZ-MYC hub [77,78,79,80]. Plant viruses are obligate parasites and their accumulation is severely affected by the growth status of the plant host. Plant development can also affect insect vectors fitness. Viral infections improve the plant’s tolerance to drought and changes in physiological traits, such as the translocation of metabolites [113,114,115]. Therefore, it is interesting to determine whether the virus can manipulate the growth–defense network, and whether the target of JA signaling will contribute to defense–growth balance, thus providing the basis for the management of insect-borne plant viruses and breeding engineered virus/insect-proof crops.

## 5. Perspectives

Staple-food crop yield losses caused by diseases and herbivores are believed to reach more than 20% of the total crop production worldwide, e.g., 30% in rice and 22.6% in maize. Plant viral diseases are one of the most important pathogens causing agricultural losses. Currently, the emergence and reemergence of insect-transmitted plant viruses in the past decades have been mainly driven by planthoppers, whiteflies, aphids and thrips [116,117,118,119,120]. We and many other groups in the world have developed several biotechnological methods to improve crop resistance against these vector-borne viruses with the expression of virus-targeting RNA interference [121,122,123]. Due to the difficulties in the further usage of these genetically modified plants in the field, the most effective method for insect-borne plant virus management is likely to control the population of insect vectors. The defense/counter-defense interplay between viruses, vectors, and their host plants is likely a consequence of co-evolution. A few mutualistic relationships between viruses and their vectors have been reported (e.g., the whitefly transmits begomoviruses and the thrip transmits bunyaviruses). These viruses have evolved a mutualistic relationship with their shared host plants, which is regarded as reciprocal cooperation between species. This phenomenon can be explained as follows: viruses, by manipulating their host plant’s defense, alter the emissions of plant volatile organic compounds, or enhance the plant’s nutritional quality to improve the attractiveness and suitability of the vector to the infected plant host, thus promoting the vector’s performance and viral spread [124,125,126]. Therefore, fully understanding the mutualistic relationships between viruses and their vectors is vital to designing interference strategies to control disease.

Meanwhile, the ecological relevance of the tripartite interactions of virus–vector–plant needs to be further understood. The highly trophic relationship between plants, herbivores, and their natural enemies are integral elements in natural and agricultural ecosystems. The predators or parasites of herbivores, herbivores and autotrophic plants form a simple consumer-resource system. The predator plays a key role in controlling the numbers of herbivores that can transmit plant viruses [127]. The predator is also affected by virus-induced alterations in plants. It has been shown that TSWV can help thrips in reducing their vulnerable period of predation [102]. Further, not only the interactions between plants and predators but also the interactions between plants and pollinator insects can be exploited by viral vectors. CMV alters the foraging behavior of bumblebees (*Bombus terrestris*) by changing the quantity and quality of volatiles emitted from infected tomato (*Solanum lycopersicum*) plants [128]. Finally, enhanced pollinator services for infected plants might cause susceptible genes to persist in plant populations, thereby ensuring that susceptible hosts will be present for the CMV, despite the fact that bumblebees are not CMV vectors. This can be seen as a ‘payback’ from the virus to its host plant.

Overall, there is selective pressure for insect-borne plant viruses to manipulate plant’s immune systems against herbivores, since the spread of such viruses is highly reliant on their insect vectors. As the most efficient defense pathway, JA signaling is targeted by various plant viruses in different ways to cooperate with their insect vectors for more efficient dispersal to new plants. Notably, these novel insights on tripartite interactions between plants, viruses and insects, and in particular the increasing knowledge on JAs and JA signaling on plant defenses and developments, will help preventing insect-borne viral diseases in the future.

## Figures and Tables

**Figure 1 viruses-12-00148-f001:**
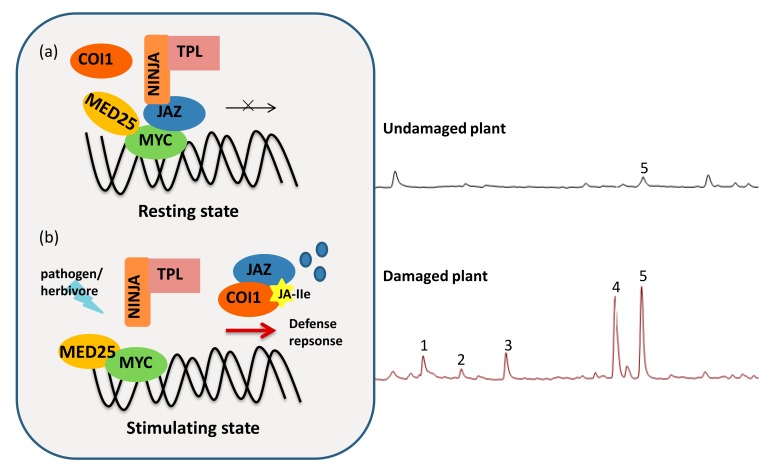
Jasmonate (JA)-mediated defense against plant biotic stresses. (**a**) In resting undamaged cells, JASMONATE ZIM (JAZ) repressor proteins bind with positive transcriptional factors, such as basic helix-loop-helix MYCs. Together with the adaptor protein NOVEL INTERACTOR OF JAZ (NINJA) and the corepressor TOPLESS (TPL) recruited by NINJA, to prevent the activation of the JA pathway. (**b**) When plants are attacked by pathogens or herbivores, the JA signaling pathway can be activated to counter the biotic stress. In a stimulated state, the F-box protein CORONATINE INSENSITIVE1 (COI1) receptor receives the jasmonic acid-isoleucine (JA-Ile) signal and then degrades the JAZ repressor through the E3 ubiquitin-ligase SKP1-Cullin-F-box complex. Once the transcriptional activators, such as MYC proteins, are released, the JA-mediated defense pathway is activated. The plant’s defensive responses will then output a series of defense responses. Plants elicit the induced chemical compounds when damaged by herbivores. The ion chromatograms shown in the right panel indicate the herbivore-induced plant volatiles (HIPVs) emitted from the headspaces of peppers (*Capsicum annuum* L.). Peppers that were infested with (lower panel) or without (upper panel) non-viruliferous western flower thrip for 24 h are also shown. The numbered peaks represent (1) 3-hexen-1-ol acetate; (2) D-limonene; (3) beta-ocimene; (4) beta-linalool; (5) 1,6-octadiene.

**Figure 2 viruses-12-00148-f002:**
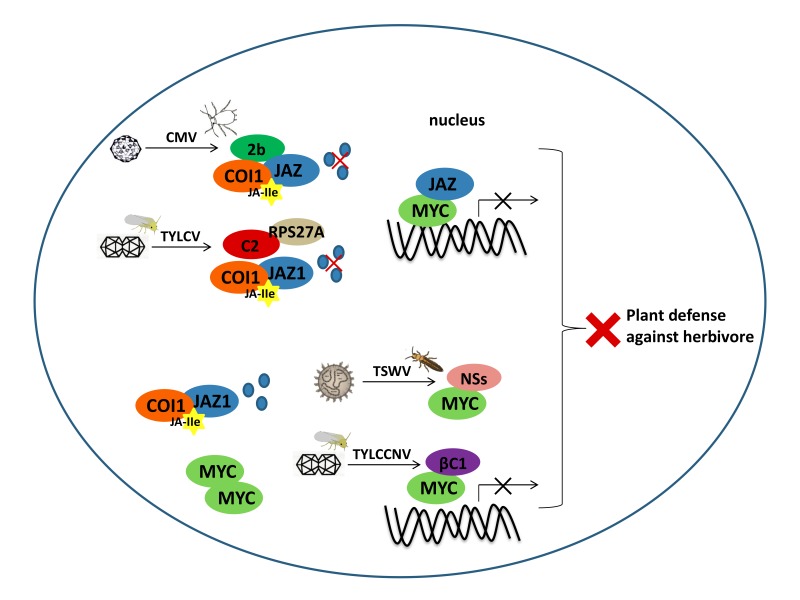
A conserved mechanism used by insect-borne plant viruses to interrupt the JAZ-MYC hub. Viral effectors attenuate the JAZ-MYC hub to interfere with plant defenses against insect vectors, thus manipulating the vector’s feeding behavior to promote viral spread. The 2b protein of the *Cucumber mosaic virus* (CMV) directly interacts with JAZ proteins and protects JAZs from degradation, thereby attenuating the JA signaling pathway to be activated. The C2 protein from the *Tomato yellow leaf curl virus* (TYLCV) interacts with plant ubiquitin RPS27A to suppress JAZ1 degradation and activation of the JA pathway. The viral genetic factor βC1 of the *Tomato yellow leaf curl China virus* (TYLCCNV) disrupts the transcriptional activity of MYC2, thus compromising activation of the MYC2-regulated chemical defense against insects. The nonstructural proteins (NSs) of orthotospoviruses manipulate JA-mediate defenses through direct interaction with MYC2. These dysfunctions of JAZ-MYC hub lead to enhanced insect vectors performance on plants.

**Table 1 viruses-12-00148-t001:** Pathogens or insects targeting the JAZ-MYC2 hub.

Species	Effectors-Plant Targets	Mechanism	Reference
**Virus**			
*Tomato yellow leaf curl China virus* (begomovirus)	βC1-MYC2	Subvert defense gene activity, compromise terpene synthase	[77]
*Tomato yellow leaf curl virus* (begomovirus)	C2-JAZ1	Compromise JAZ1 degradation, inhibit downstream gene regulated defense	[78]
*Tomato spotted wilt orthotospovirus* (tospovirus)	NSs-MYC2	Directly interact with MYCs to disable JA-mediated host defenses against the thrip vector	[79]
*Cucumber mosaic virus* (bromovirus)	2b-JAZ	Repress the JA-induced degradation of JAZ proteins	[80]
**Bacterium and Fungal**			
*Pseudomonas syringae pv. tabaci (Pta) 11528*	HopX1-JAZ	Encode a cysteine protease to promote the degradation of JAZ proteins	[81]
*Pseudomonas syringae* strain A2	HopZ1a-JAZ	JAZs can be acetylated by HopZ1a through putative acetyltransferase activity.	[82]
*Pseudomonas syringae*	AvrB-JAZ	Induce the COI1–JAZs interactions and the degradation of multiple JAZ proteins	[83]
*Laccaria bicolor*	MiSSP7-JAZ	Interact with and stabilize JAZ proteins against JA-Ile mediated degradation	[84]
*Pseudomonas syringae*	HopBB1-JAZ	Utilize JAZ3 to target TCP14 to the SCFCOI1 degradation complex	[85]
*Pseudomonas fluorescens WCS417r*	? /-MYC2	Utilize MYC2 to enhance JA-mediated induced systemic resistance against pathogens and insect herbivores	[86]
*Pseudomonas syringae*	COR-COI1/JAZ	Depredate JAZ, activate the host’s JA signaling pathway to suppress salicylic acid to promote bacterial virulence	[87]
**Insect**			
*Helicoverpa armigera*	HARP1-JAZ	Directly interact with JAZ to prevent its COI1-mediated degradation	[70]
*Pieris brassicae*	egg extract-MYC	Diminished MYC protein levels in a SA-dependent manner	[57]

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
