# Peer review of "Manipulation of Jasmonate Signaling by Plant Viruses and Their Insect Vectors"

_viruses, 2020, doi:10.3390/v12020148_

Round 1

Reviewer 1 Report

Comments to the authors

This review comprises an overview on how jasmonate signaling in plants can be manipulated by insect-borne plant viruses and their vectors.

Overall the authors have done a good job to summarise current findings in the field. Some grammatical edits and corrections are needed. In particular, towards the end of the review, the authors need a stronger closing paragraph to get their points across – please see comments below. Often times, due to grammatical error, it is difficult to understand what the authors are trying to convey.

My first comment is whether the title can perhaps be made more concise ? ‘Manipulation of jasmonate signaling by plant viruses and their insect vectors’ can be an option. Because the review not only covered how jasmonate signaling can be manipulated by insect vectors, but also by insect herbivores (not vectors of any plant viruses). Therefore a more general title may benefit, to be more representative of what the review comprises.

Also in the overall text, please be careful on the usage of the words of ‘insect-borne viruses’ or ‘insect-borne plant viruses’ as those two have very different meanings.

Comments on the manuscript itself

Line 11 : Change ‘insect-borne viruses’ to ‘insect-borne plant viruses’

Line 11-12 : Rewrite the sentence to make it more concise, i.e : “…serious threats to stable crop yield, mainly due to causing severe plant disease ..” are repetitive.  It can be re written to : “..serious threats to a reduction in crop yield and causing secondary damage to the host plant from herbivory..”.

Line 16 : Change ‘to survival’ to ‘For survival’

Line 17 : Change ‘convergent’ to ‘convergently’ ?

Line 19 : ‘vector borne plant viral diseases’

Line 20-21 : Shorten the sentence – i.e : ..to further design and develop effective strategies..

Line 32 : ‘persistently’

Line 35 : delete ‘different’

Line 36 : change feeding orientation to ‘feeding preference’

Line 36 : ‘persistently’. Several times the authors used the word ‘persistent’ whilst it should be ‘persistently’. Please check.

Line 40 : ‘initial settlement’, and ‘later preference’ or delayed preference. Please re-check grammar and sentence.

Line 45 : can add Westwood et al. 2010 whereby Cucumber mosaic virus infection on Arabidopsis increases the amount of aphid feeding deterrent glucosinolates, hence reducing aphid fitness.

Line 45 : ‘promote’ or ‘enhance’

Line 54 : participate

Line 60 : delete ‘a kind of’

Line 63 : delete ‘inside’

Line 93 : fine tuning

Line 93-94 : Unclear sentence here where the author states that JA could ‘confer versatile and adaptive traits’ to plants between defense and development. Can the authors add more references  to back up this sentence please? Pacheco et al. (2012) on its own is not sufficient. Also Pacheco et al. (2012) did not directly states that JA is involved in plant development. If any, their work highlights more the role that JA plays in antiviral defence in Nicotiana benthamiana.

Line 98 : to amplify, not ‘amply’

Line 99-100 : add Howe and Jander, 2008 as reference for the role of glucosinolates against aphids

Line 104 : Can the authors explain in more details on how these extraneous JA application reduced the infection in these two studies being referenced here? Perhaps just in a sentence or two.

Line 106-108 : This last sentence does not quite follow the sentence beforehand where the authors mentioned about VOCs. Can the authros rewrite this sentence , or extend the section about the VOCs ? What effects does changes in VOCs causes to the insect vectors for example? Whether it is quantitative or qualitative changes in VOCs?

Figure 1 : The ion chromatograms, whilst informative, have no reference to where it was taken from. Can the authors add the reference please? And what kind of herbivory causes the changes in the chromatograms? i.e : by which insect, or whether its by an insect vector carrying a plant virus.

Line 123 : This sentence is a bit unclear, perhaps rewrite to : The counter-defense of virus and herbivore to JA signaling in plant ?

Line 125 : plants ‘are’ equipped.

Line 128 : by strongly diminishing

Line 135 : Can the authors add the works by Saskia Hogenhout et al regarding aphid salivary effectors in this paragraph?

Line 159-160 : This first sentence is a very broad generalization. Perhaps delete this first sentence and start the paragraph immediately with “Insect vectors can get benefits by transmitting plant pathogens that help them to suppress….”

Line 161-162 : Caution needed when making a statement that ‘JAs and JA signallling mediate antiviral defenses in plants’ as JA are more commonly known to mediate defenses against insect pathogen and herbivores. Rewrite this sentence perhaps?

Line 164 : Delete ‘the’

Line 166 : Change ‘JA-mediated’ to ‘ JA mediates’

Line 168-169 : This sentence is repetitive, as it carries the same message as the sentence directly after it at L. 169-171.

Line 172 : Add references for this statement.

Line 173 : previously

Line 174 : delete ‘genetic’

Line 179 : targeting

Line 182 : delete ‘of’

Line 194 : change to ‘Beside plant viruses, bacterial pathogen..’

Line 197 : Unclear on how this can be a consequence of coevolution between pathogen and plants? Can the author expand on this sentence?

Line 201-202 : This particular sentence needs to be re-written as its meaning is unclear. Correction in grammar and the use of English in this sentence is needed.

Line 205 : convergently

Line 221 : ..’..a virus can target’

Line 224 : reference?

Line 227-228 : Can this sentence be made shorter or split into two? I understood what the authors are saying but it became confusing as the sentence is too long.

Line 239 : change ‘leading’ to ‘leads’

Line 248 – 249 : Can the authors add more references to which showed that plant viruses changes volatile emission, and this may or may not always affect aphid settling preferences? i.e : Mauck et al., 2010; Tungadi et al., 2019.

Line 256 : persistentetly

Line 260 : non-persistently

Line 261 : beside Mauck et al., the works by Westwood et al., 2010 also showed this in the CMV-Arabidopsis-aphid pathosystem.

Line 262 : Although interesting, but how does this plant defense-growth trade-off regulated by JA linked to manipulation of JA signaling by insect vectors? Can the authors write a few sentences to expand this section and how it can link to , for example, increased performances to insect vectors?

Line 268 : fine-tunes

Line 270 : change ‘are interacted’ to ‘interacts’

Line 279-280 : Can the authors add more references to how virus infection can improve the host plant’s tolerance to drought? i.e : Westwood et al, 2013 whereby CMV infection renders Arabidopsis thaliana to become more drought tolerant.

Line 285 : What about aphids?

Line 287-288 : Can the authors add more references to back up this statement? i.e : Worrall et al. 2019 showed that exogenous application of RNAi-inducing dsRNA can reduce aphid-mediated transmission of a plant virus.

Line 290 : genetically modified plants

Line 296-297 : manipulating, enhancing and altering. Please check the grammar.

Line 299-300 : Unclear sentence, does the authors mean highly efficient tranmissions?

Line 300-301 : Same comment as above, it is hard to understand the meaning of this sentence. Can the authors rewrite this section? How does ‘fully understanding plant-dependent indirect manipulation by virus on their insect vector” can ‘break mutualistic relationship of viruses and their vectors” ? Unclear sentence and confusing to understand what the author is trying to say in this section.

Line 303 : delete ‘detailed’

Line 307 : More references needed to back up this claim of ‘it is also affected by virus-induced alterations in plant headspace VOCs’

Line 313 : Indeed, can the authors expand on how this may benefit CMV? i.e : to ensure that susceptible genes persist in plant populations, hence ensuring that susceptible hosts will be present to CMV. Despite the fact that bumblebees are not CMV vectors. Therefore this can be seen as a ‘payback’ from the virus to its host plant.

Can the authors add a sentence or two to end this review by summarizing on how understanding the different ways that plant viruses can manipulate JA signaling , how can this be utilised help to control the spread of plant virus diseases ? How does this fit in the wider ecological perspective?

Also does the authors find any references on how manipulation of this JA signaling affects natural enemies or predators of the insect vectors?

Lastly, how does manipulation of JA signaling can benefit virus transmission? Can the authros summarise that too at the end?

Overall this is a nicely thought of and well written review. However there are parts where it lacks coherence and the sentences are difficult to understand, due to poor grammar and poor English. More editing is needed to make the sentences clearer and more consice. Authors would also need to add more references in some parts of the manuscripts, please see comments above.

Author Response

Response to Reviewer 1 Comments

Point 1: My first comment is whether the title can perhaps be made more concise? ‘Manipulation of jasmonate signalling by plant viruses and their insect vectors’ can be an option. Because the review not only covered how jasmonate signalling can be manipulated by insect vectors, but also by insect herbivores (not vectors of any plant viruses). Therefore a more general title may benefit, to be more representative of what the review comprises.

Response 1: Thanks for the kind suggestion. We have changed the title as you suggested. Besides, we also made the corresponding changes on the whole manuscript according to this title change.

Point 2: Line 11: Change ‘insect-borne viruses’ to ‘insect-borne plant viruses’

Response 2: Done.

Point 3: Line 11-12: Rewrite the sentence to make it more concise, i.e : “…serious threats to stable crop yield, mainly due to causing severe plant disease ..” are repetitive.  It can be re written to : “..serious threats to a reduction in crop yield and causing secondary damage to the host plant from herbivory..”.

Response 3: We have fixed the sentences as suggested.

Point 4: Line 16: Change ‘to survival’ to ‘For survival’

Response 4: Done.

Point 5: Line 17: Change ‘convergent’ to ‘convergently’?

Response 5: Done.

Point 6: Line 19: ‘vector borne plant viral diseases’

Response 6: Edited.

Point 7: Line 20-21: Shorten the sentence – i.e : ..to further design and develop effective strategies..

Response 7: Edited.

Point 8: Line 32: ‘persistently’

Response 8: Edited.

Point 9: Line 35: delete ‘different’

Response 9: Done.

Point 10: Line 36: change feeding orientation to ‘feeding preference’

Response 10: Done.

Point 11: Line 40: ‘initial settlement’, and ‘later preference’ or delayed preference. Please re-check grammar and sentence.

Response 11: Thanks for the comments. We have edited the sentences as follows: which indicates an initial settlement on infected plants and a later preference for mock-inoculated plants.

Point 12: Line 45: can add Westwood et al. 2010 whereby Cucumber mosaic virus infection on Arabidopsis increases the amount of aphid feeding deterrent glucosinolates, hence reducing aphid fitness.

Response 12: Thanks for the suggestion. We have added the recommended reference.

Point 13: Line 45: ‘promote’ or ‘enhance’

Response 13: Edited.

Point 14: Line 54: participate

Response 14: Edited.

Point 15: Line 60: delete ‘a kind of’

Response 15: Done.

Point 16: Line 63: delete ‘inside’

Response 16: Done.

Point 17: Line 93: fine tuning

Response 17: Edited.

Point 18: Line 93-94: Unclear sentence here where the author states that JA could ‘confer versatile and adaptive traits’ to plants between defense and development. Can the authors add more references to back up this sentence please? Pacheco et al. (2012) on its own is not sufficient. Also Pacheco et al. (2012) did not directly states that JA is involved in plant development. If any, their work highlights more the role that JA plays in antiviral defence in Nicotiana benthamiana.

Response 18: Sorry for the confusion. We have added more reference to support this statement (Zhou, W. et al. 2019. DOI: 10.1016/j.cell.2019.03.006.). Certainly, Pacheco et al. (2012) puts more emphasis on the role of JA in antiviral defences. We concluded the JA role in development based on the report that silencing the jasmonate perception gene accelerated virus caused cell death symptom which serious affect plant development. 

Point 19: Line 98: to amplify, not ‘amply’

Response 19: Fixed.

Point 20: Line 99-100: add Howe and Jander, 2008 as reference for the role of glucosinolates against aphids

Response 20: Added.

Point 21: Line 104: Can the authors explain in more details on how these extraneous JA application reduced the infection in these two studies being referenced here? Perhaps just in a sentence or two.

Response 21: We have added detailed explanation about this point.

Point 22: Line 106-108: This last sentence does not quite follow the sentence beforehand where the authors mentioned about VOCs. Can the authros rewrite this sentence, or extend the section about the VOCs? What effects does changes in VOCs causes to the insect vectors for example? Whether it is quantitative or qualitative changes in VOCs?

Response 22: We have rewritten the sentences, and we discussed the function of VOCs in more detail in the following text.

Point 23: Figure 1: The ion chromatograms, whilst informative, have no reference to where it was taken from. Can the authors add the reference please? And what kind of herbivory causes the changes in the chromatograms? i.e : by which insect, or whether its by an insect vector carrying a plant virus.

Response 23: Thanks for your suggestion. The Gas chromatograms are taken from our own results, similar results could be found from our recent publication Wu et al. PLoS Pathogens 2019. We have clarified this on the figure legend in the revised version of manuscript.

Point 24: Line 123: This sentence is a bit unclear, perhaps rewrite to: The counter-defense of virus and herbivore to JA signaling in plant?

Response 24: Fixed.

Point 25: Line 125: plants ‘are’ equipped.

Response 25: Fixed.

Point 26: Line 128: by strongly diminishing

Response 26: Fixed.

Point 27: Line 135: Can the authors add the works by Saskia Hogenhout et al regarding aphid salivary effectors in this paragraph?

Response 27: Thanks for the suggestion. We have added the reference.

Point 28: Line 159-160: This first sentence is a very broad generalization. Perhaps delete this first sentence and start the paragraph immediately with “Insect vectors can get benefits by transmitting plant pathogens that help them to suppress….”

Response 28: We have deleted the sentence as suggested.

Point 29: Line 161-162: Caution needed when making a statement that ‘JAs and JA signalling mediate antiviral defenses in plants’ as JA are more commonly known to mediate defenses against insect pathogen and herbivores. Rewrite this sentence perhaps?

Response 29: Thanks for the suggestion. We have rewritten the sentence as: Besides their well-known roles in pathogen and herbivore defense, JAs and JA signaling mediate antiviral defenses in plants.

Point 30: Line 164: Delete ‘the’

Response 30: Done.

Point 31: Line 166: Change ‘JA-mediated’ to ‘JA mediates’

Response 31: Fixed.

Point 32: Line 168-169: This sentence is repetitive, as it carries the same message as the sentence directly after it at L. 169-171.

Response 32: We have fixed it by integrating the sentences.

Point 33: Line 172: Add references for this statement.

Response 33: Added.

Point 34: Line 173: previously

Response 34: Fixed.

Point 35: Line 174: delete ‘genetic’

Response 35: Done.

Point 36: Line 179: targeting

Response 36: Done.

Point 37: Line 182: delete ‘of’

Response 37: Fixed.

Point 38: Line 194: change to ‘Beside plant viruses, bacterial pathogen..’

Response 38: Fixed.

Point 39: Line 197: Unclear on how this can be a consequence of coevolution between pathogen and plants? Can the author expand on this sentence?

Response 39: We have edited the sentences. As for pathogens, the highly evolved plant defenses seem to be a selective pressure. Pathogens which can overcome the defenses will establish successfully infection. So we regarded this as a consequence of adaptive traits evolved by pathogens to counter plant host defense systems.

Point 40: Line 201-202: This particular sentence needs to be re-written as its meaning is unclear. Correction in grammar and the use of English in this sentence is needed.

Response 40: We have edited the sentence as follows: A previous study analyzed the evolution of more than 1000 JAZ sequence proteins used bioinformatics showing highly conserved features along the evolutionary scale.

Point 41: Line 205: convergently

Response 41: Fixed.

Point 42: Line 221: ..’..a virus can target’

Response 42: Fixed.

Point 43: Line 224: reference?

Response 43: Thanks for the comments. We have already added the reference.

Point 44: Line 227-228: Can this sentence be made shorter or split into two? I understood what the authors are saying but it became confusing as the sentence is too long.

Response 44: We have split into two sentences to make it clear.

Point 45: Line 239: change ‘leading’ to ‘leads’

Response 45: Fixed.

Point 46: Line 248 – 249: Can the authors add more references to which showed that plant viruses changes volatile emission, and this may or may not always affect aphid settling preferences? i.e : Mauck et al., 2010; Tungadi et al., 2019.

Response 46: Added.

Point 47: Line 256: persistentetly

Response 47: Fixed.

Point 48: Line 260: non-persistently

Response 48: Fixed.

Point 49: Line 261: beside Mauck et al., the works by Westwood et al., 2010 also showed this in the CMV-Arabidopsis-aphid pathosystem.

Response 49: We have added the reference.

Point 50: Line 262: Although interesting, but how does this plant defense-growth trade-off regulated by JA linked to manipulation of JA signaling by insect vectors? Can the authors write a few sentences to expand this section and how it can link to, for example, increased performances to insect vectors?

Response 50: That’s an interesting issue. There are no related reports to date, "increased performances to insect vectors” can be a possibility. Defense and growth are considered to be two antagonistic processes. Insect manipulated JA to promote growth and attenuated defense, thereby enhance their performance.

Point 51: Line 268: fine-tunes

Response 51: Fixed.

Point 52: Line 270: change ‘are interacted’ to ‘interacts’

Response 52: Done.

Point 53: Line 279-280: Can the authors add more references to how virus infection can improve the host plant’s tolerance to drought? i.e : Westwood et al, 2013 whereby CMV infection renders Arabidopsis thaliana to become more drought tolerant.

Response 53: Thanks. We have added the reference.

Point 54: Line 285: What about aphids?

Response 54: Thanks for your comments. We have added aphids in.

Point 55: Line 287-288: Can the authors add more references to back up this statement? i.e : Worrall et al. 2019 showed that exogenous application of RNAi-inducing dsRNA can reduce aphid-mediated transmission of a plant virus.

Response 55: We have added the reference.

Point 56: Line 290: genetically modified plants

Response 56: Fixed.

Point 57: Line 296-297: manipulating, enhancing and altering. Please check the grammar.

Response 57: Edited.

Point 58: Line 299-300: Unclear sentence, does the authors mean highly efficient tranmissions?

Response 58: Yes, we have edited the sentence.

Point 59: Line 300-301: Same comment as above, it is hard to understand the meaning of this sentence. Can the authors rewrite this section? How does ‘fully understanding plant-dependent indirect manipulation by virus on their insect vector” can ‘break mutualistic relationship of viruses and their vectors” ? Unclear sentence and confusing to understand what the author is trying to say in this section.

Response 59: Sorry for the confusion. We have edited the sentences. We planned to express as: based on the theoretical research in virus-plant-insect interactions, we can develop interference strategies to break mutualistic relationship between them.

Point 60: Line 303: delete ‘detailed’

Response 60: Done.

Point 61: Line 307: More references needed to back up this claim of ‘it is also affected by virus-induced alterations in plant headspace VOCs’

Response 61: It is one of our hypotheses to explain some data from an on-going project. We have turned down this statement. We will make explicit this concept in another manuscript.

Point 62: Line 313: Indeed, can the authors expand on how this may benefit CMV? i.e : to ensure that susceptible genes persist in plant populations, hence ensuring that susceptible hosts will be present to CMV. Despite the fact that bumblebees are not CMV vectors. Therefore this can be seen as a ‘payback’ from the virus to its host plant.

Response 62: Thanks for the kind suggestion. We have described this issue more detail.

Point 63: Can the authors add a sentence or two to end this review by summarizing on how understanding the different ways that plant viruses can manipulate JA signaling, how can this be utilised help to control the spread of plant virus diseases? How does this fit in the wider ecological perspective? Also does the authors find any references on how manipulation of this JA signaling affects natural enemies or predators of the insect vectors? Lastly, how does manipulation of JA signaling can benefit virus transmission? Can the authors summarise that too at the end?

Response 63: Thanks for the suggestion. We have added some sentence in this part to make them presented in a better logic way. Firstly, we have added a summary paragraph to summary this part. Next, insect-borne plant viruses manipulated JA to promote transmission mainly in two ways. Indirectly, virus attenuated JA mediated resistance to insects, thereby promote insect vectors’ performance. Directly, JA has effects on viral defenses too. However, whether JA manipulation directly benefit to virus accumulation need to be future determined. We have summarized this part in section 2 and 3.2.

Reviewer 2 Report

The manuscript by Wu and Ye addresses an original topic, that has not been frequently reviewed, at least for the the virus manipulation part.

The manuscript is generally well organized, but suffers writing problems in different sections, such as: line 54, 94, 98, 102-103, 106, 126, 135, 138, 296, 303…

Some defense subversion mechanisms could be defined in more details.

GS for glucosinolates should be defined at 1st occurrence (line 47)

Figure 1 does not indicate the origin of the chromatogram.

Line 198 : should precise that Mp has only one JAZ gene.

Two more papers could be cited :

  Carr et al (2018) Adv Virus Res 102, 177. Tungadi et al (2019) Mol Plant Pathol doi: 10.1111/mpp.12892

All references have only 2 authors…

Author Response

Response to Reviewer 2 Comments

Point 1: The manuscript is generally well organized, but suffers writing problems in different sections, such as: line 54, 94, 98, 102-103, 106, 126, 135, 138, 296, 303ɉ۬

Response 1: We have corrected these writing problems.

Point 2: GS for glucosinolates should be defined at 1st occurrence (line 47)

Response 2: Thanks for the comments. We have fixed it.

Point 3: Figure 1 does not indicate the origin of the chromatogram.

Response 3: Thanks for the comments. The Gas chromatograms are the headspace compounds of peppers (Capsicum annuum L.) which are infested with western flower thrip for 24h or mock control. We have updated this description in the figure legend.

Point 4: Line 198: should precise that Mp has only one JAZ gene.

Response 4: We have edited the statement as you suggested.

Point 5: Two more papers could be cited: Carr et al (2018) Adv Virus Res 102, 177. Tungadi et al (2019) Mol Plant Pathol doi: 10.1111/mpp.12892

All references have only 2 authors…

Response 5: Thanks for the comments. We have added these suggested references, and the problem of reference format has rectified.

Reviewer 3 Report

This is a wonderful useful Review, but just need some english modifications to be more clear
line 12: plant diseases and second damage from herbivore-vector’s infestation. you need to choose another word instead of "SECOND"
Line 21 :sustainable strategies to protect viral damages so as to increase crop yield and food security. Modify the sentence structure
Line 25: There are huge numbers and varieties of plant viruses. word "Varieties" should not be for microorganisms, choose another term.
Line 34: gene expression alteration as reviewed before.Modify the sentence structure
Line 54: In this review, we will summarize how JAs participant. I think you mean Participate
Line 159: Compared to millions of species of insects, only small amount. Better to choose another word instead of small amount
Line 246: abundant ingredients in plant volatiles. It will better if you used the term"compound" instead of "Ingredients"

Author Response

Response to Reviewer 3 Comments

Point 1: line 12: plant diseases and second damage from herbivore-vector’s infestation. you need to choose another word instead of "SECOND"

Response 1: We have changed the word to “secondary”.

Point 2: Line 21: sustainable strategies to protect viral damages so as to increase crop yield and food security. Modify the sentence structure

Response 2: Edited.

Point 3: Line 25: There are huge numbers and varieties of plant viruses. word "Varieties" should not be for microorganisms, choose another term.

Response 3: Thanks for the comments. We rewrite the phrase “huge numbers and varieties of” to “many” for more concise.

Point 4: Line 34: gene expression alteration as reviewed before. Modify the sentence structure

Response 4: Edited.

Point 5: Line 54: In this review, we will summarize how JAs participant. I think you mean Participate

Response 5: Yes. We have fixed this mistake.

Point 6: Line 159: Compared to millions of species of insects, only small amount. Better to choose another word instead of small amount

Response 6: Thanks for the comments. We have deleted the sentences.

Point 7: Line 246: abundant ingredients in plant volatiles. It will better if you used the term "compound" instead of "Ingredients"

Response 7: Thanks for the suggestion. We have changed the word as you suggested.

This manuscript is a resubmission of an earlier submission. The following is a list of the peer review reports and author responses from that submission.

Round 1

Reviewer 1 Report

The review presented by KWu and Ye  is focused in the role that play the jasmonate response in a three way interaction: viruses-plant-insect. This is indeed a really relevant topic for plant pathology, however despite of a tantalising title,  the review is disappointed.

A large number of the selected references are inadequate. The text lack of a proper structure. and the ideas and messages are mixed up making the reading of the manuscript a hard and frustrated exercise. Besides the  manuscript requires a large and profound grammar revision .

Taking together all the comments, I consider that the manuscript  requires a substantial review to be published in  viruses.

Reviewer 2 Report

at current state I cannot provide a proper revision for this MS. The English language is precarious, hard to read and/or miss leading. However, I believe this is an interesting review which could improve both the field of Jasmonates and plant Virus research. Maybe you can ask for editorial assistance and resubmit the work.